# Learning Signal-Agnostic Manifolds of Neural Fields

**Yilun Du**[1]
yilundu@mit.edu

**Katherine Collins**[1,2]
katiemc@mit.edu

**Joshua B. Tenenbaum**[1,2,3]
jbt@mit.edu

**Vincent Sitzmann**[1]
sitzmann@mit.edu

[1]MIT CSAIL    [2]MIT BCS    [3] MIT CBMM
https://yilundu.github.io/gem/

## Abstract

Deep neural networks have been used widely to learn the latent structure of datasets, across modalities such as images, shapes, and audio signals. However, existing models are generally modality-dependent, requiring custom architectures and objectives to process different classes of signals. We leverage neural fields to capture the underlying structure in image, shape, audio and cross-modal audiovisual domains in a *modality-independent* manner. We cast our task as one of learning a manifold, where we aim to infer a low-dimensional, locally linear subspace in which our data resides. By enforcing coverage of the manifold, local linearity, and local isometry, our model — dubbed GEM — learns to capture the underlying structure of datasets across modalities. We can then travel along linear regions of our manifold to obtain perceptually consistent interpolations between samples, and can further use GEM to recover points on our manifold and glean not only diverse completions of input images, but cross-modal hallucinations of audio or image signals. Finally, we show that by walking across the underlying manifold of GEM, we may generate new samples in our signal domains[1].

## 1 Introduction

Every moment, we receive and perceive high-dimensional signals from the world around us. These signals are in constant flux; yet remarkably, our perception system is largely invariant to these changes, allowing us to efficiently infer the presence of coherent objects and entities across time. One hypothesis for how we achieve this invariance is that we infer the underlying manifold in which perceptual inputs lie [1], naturally enabling us to link high-dimensional perceptual changes with local movements along such a manifold. In this paper, we study how we may learn and discover a low-dimensional manifold in a *signal-agnostic* manner, over arbitrary perceptual inputs.

Manifolds are characterized by three core properties [2, 3]. First, a manifold should exhibit data coverage, i.e., all instances and variations of a signal are explained in the underlying low-dimensional space. Second, a manifold should be locally metric, enabling perceptual manipulation of a signal by moving around the surrounding low-dimensional space. Finally, the underlying structure of a manifold should be globally-consistent; e.g. similar signals should be embedded close to one another.

Existing approaches to learning generative models, such as GANs [4], can be viewed as instances of manifold learning. However, such approaches have two key limitations. First, low-dimensional latent codes learned by generative models do not satisfy all desired properties for a manifold; while the underlying latent space of a GAN enables us to perceptually manipulate a signal, GANs suffer from mode collapse, and the underlying latent space does not cover the entire data distribution. Second, existing generative architectures are biased towards particular signal modalities, requiring custom architectures and losses depending on the domain upon which they are applied – thereby preventing us from discovering a manifold across arbitrary perceptual inputs in a signal-agnostic manner.

---

[1]Code and additional results are available at https://yilundu.github.io/gem/.

35th Conference on Neural Information Processing Systems (NeurIPS 2021).

As an example, while existing generative models of images are regularly based on convolutional neural networks, the same architecture in 1D does not readily afford high-quality modeling of audio signals. Rather, generative models for different domains require significant architecture adaptations and tuning. Existing generative models are further constrained by the common assumption that training data lie on a regular grid, such as grids of pixels for images, or grids of amplitudes for audio signals. As a result, they require uniformly sampled data, precluding them from adequately modeling irregularly sampled data, like point clouds. Such a challenge is especially prominent in cross-modal generative modeling, where a system must jointly learn a generative model over multiple signal sources (e.g., over images and associated audio snippets). While this task is trivial for humans – we can readily recall not only an image of an instrument, but also a notion of its timbre and volume – existing machine learning models struggle to jointly fit such signals without customization.

To obtain *signal-agnostic* learning of manifolds over signals, we propose to model distributions of signals in the function space of neural fields, which are capable of parameterizing image, audio, shape, and audiovisual signals in a modality-independent manner. We then utilize hypernetworks [5], to regress individual neural fields from an underlying latent space to represent a signal distribution. To further ensure that our distribution over signals corresponds to a manifold, we formulate our learning objective with explicit losses to encourage the three desired properties for a manifold: data coverage, local linearity, and global consistency. The resulting model, which we dub GEM[2], enables us to capture the manifold of a variety of signals – ranging from audio to images and 3D shapes – with almost no architectural modification, as illustrated in Figure 1. We further demonstrate that our approach reliably recovers distributions over cross-modal signals, such as images with a correlated audio snippet, while also eliciting sample diversity.

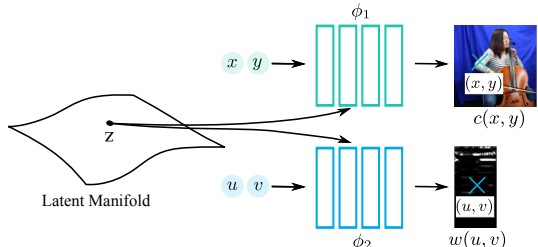

Figure 1: GEM learns a low-dimensional latent manifold over signals. Given a cross-modal signal, latents in GEM are mapped, using a hypernetwork $\psi$, into neural networks $\phi_1$ and $\phi_2$. $\phi_1$ represents a image by mapping each pixel position $(x, y)$ to its associated color $c(x, y)$. $\phi_2$ represents an audio spectrogram by mapping each pixel position $(u, v)$ to its intensity $w(u, v)$. This enables GEM to be applied in a domain agnostic manner across separate (multi-modal) signals, by utilizing a separate function $\phi$ for each mode of a signal.

We contribute the following: first, we present GEM, which we show can learn manifolds over images, 3D shapes, audio, and cross-modal audiovisual signals in a signal-agnostic manner. Second, we demonstrate that our model recovers the global structure of each signal domain, permitting easy interpolation between nearby signals, as well as completion of partial inputs. Finally, we show that walking along our learned manifold enables us to generate new samples for each modality.

## 2 Related Work

**Manifold Learning.** Manifold learning is a large and well-studied topic [2, 3, 6–10] which seeks to obtain the underlying non-linear, low-dimensional subspace in which naturally high-dimensional input signals lie. Many early works in manifold learning utilize a nearest neighbor graph to obtain such a low-dimensional subspace. For instance, Tenenbaum *et al.* [2] employs the geodesic distance between nearby points, while Roweis *et al.* [3] use locally linear subregions around each subspace. Subsequent work has explored additional objectives [7–10] to obtain underlying non-linear, low-dimensional subspaces. Recently, [11–13] propose to combine traditional manifold learning algorithms with deep networks by training autoencoders to explicitly encourage latents to match embeddings obtained from classical manifold learning methods. Further, [14] propose to utilize a heat kernel diffusion process to uncover the underlying manifold of data. In this work, we show how we may combine insights from earlier works in manifold learning with modern deep learning techniques to learn continuous manifolds of data in high-dimensional signal spaces. Drawing on these two traditions enables our model to smoothly interpolate between samples, restore partial input signals, and further generate new data samples in high-dimensional signal space.

---

[2]short for GEnerative Manifold learning

**Learning Distributions of Neural Fields.**    Recent work has demonstrated the potential of treating fully connected networks as continuous, memory-efficient implicit (or coordinate-based) representations for shape parts [15, 16], objects [17–20], or scenes [21–24]. Sitzmann et al. [25] showed that these coordinate-based representations may be leveraged for modeling a wide array of signals, such as audio, video, images, and solutions to partial differential equations. These representations may be conditioned on auxiliary input via conditioning by concatenation [17, 26], hypernetworks [21], gradient-based meta-learning [27], or activation modulation [28]. Subsequently, they may be embedded in a generative adversarial [28, 29] or auto-encoder-based framework [21]. However, both of these approaches require modality-dependent architecture choices. Specifically, modification is necessary in the encoder of auto-encoding frameworks, or the discriminator in adversarial frameworks. In contrast, we propose to represent the problem of learning a distribution as one of learning the underlying *manifold* in which the signals lie, and show that this enables signal-agnostic modeling. Concurrent to our work, Dupont et al. [30] propose a signal-agnostic discriminator based on pointwise convolutions, which enables discrimination of signals that do not lie on regular grids. In contrast, we do not take an adversarial approach, thereby avoiding the need for a convolution-based discriminator. We further show that our learned manifold better captures the underlying data manifold and demonstrate that our approach is able to capture manifolds across cross-modal distributions, which we find destabilizes training of [30].

**Generative Modeling.**    Our work is also related to existing work in generative modeling. GANs [4] are a popular framework used to model modalities such as images [31] or audio [32], but often suffer from training instability and mode collapse. VAEs [33], another class of generative models, utilize an encoder and decoder to learn a shared latent space for data samples. Recently autoregressive models have also been used for image [34], audio [35], and multi-modal text and image generation [36]. In contrast to previous generative approaches, which require custom architectures for encoding and decoding, our approach is a *modality-independent* method for generating samples distribution and presents a new view of generative modeling: one of learning an underlying manifold of data.

## 3    Parameterizing Spaces of Signals with Neural Fields

An arbitrary signal $\mathcal{I}$ in $\mathbb{R}^{D_1 \times \cdots \times D_k}$ can be represented efficiently as a function $\Phi$ which maps each coordinate $\boldsymbol{x} \in \mathbb{R}^k$ to the value $\boldsymbol{v}$ of the feature coordinate at that dimension.

$$\Phi : \mathbb{R}^k \to \mathbb{R}, \quad \boldsymbol{x} \to \Phi(\boldsymbol{x}) = \boldsymbol{v} \tag{1}$$

Such a formulation enables us to represent any signal, such as an image, shape, or waveform – or a set of signals, like images with waveforms – as a continuous function $\Phi$. We use a three-layer multilayer perceptron (MLP) with hidden dimension 512 as our $\Phi$. Only the input dimension of $\Phi$ is varied, depending on the dimensionality of the underlying signal – all other architectural choices are the same across signal types. We refer to $\Phi$ as a neural field.

**Parameterizing Spaces over Signals.**    A set of signals may be parameterized as a subspace $\mathcal{F}$ of functions $\Phi_i$. To efficiently parameterize functions $\Phi$, we represent each $\Phi$ using a low-dimensional latent space in $\mathbb{R}^n$ and map latents to $\mathcal{F}$ via a hypernetwork $\Psi$:

$$\Psi : \mathbb{R}^n \to \mathcal{F}, \quad \boldsymbol{z} \to \Psi(\boldsymbol{z}) = \Phi, \tag{2}$$

To regress $\Phi$ with our hypernetwork, we predict individual weights $\boldsymbol{W}^{\mathcal{L}}$ and biases $\boldsymbol{b}^{\mathcal{L}}$ for each layer $\mathcal{L}$ in $\Phi$. Across all experiments, our hypernetwork is parameterized as a three-layer MLP (similar to our $\Phi$), where all weights and biases of the signal representation $\Phi$ are predicted in the final layer. Directly predicting weights $\boldsymbol{W}^{\mathcal{L}}$ is a high-dimensional regression problem, so following [37], we parameterize $\boldsymbol{W}$ via a low-rank decomposition as $\boldsymbol{W}^{\mathcal{L}} = \boldsymbol{W}_s^{\mathcal{L}} \odot \sigma(\boldsymbol{W}_h^{\mathcal{L}})$, where $\sigma$ is the sigmoid function. $\boldsymbol{W}_s^{\mathcal{L}}$ is shared across all predicted $\Phi$, while $\boldsymbol{W}_h^{\mathcal{L}} = A^{\mathcal{L}} \times B^{\mathcal{L}}$ is represented as two low-rank matrices $A^{\mathcal{L}}$ and $B^{\mathcal{L}}$ – and is regressed separately for each individual latent $\boldsymbol{z}$.

## 4    Learning Structured Manifold of Signals

Given a training set $\mathcal{C} = \{\mathcal{I}_i\}_{i=1}^N$, of $N$ distinct arbitrary signals $\mathcal{I}_i \in \mathbb{R}^{D_1 \times \cdots \times D_k}$, our goal is to learn – in a *signal-agnostic* manner – the underlying low-dimensional *manifold* $\mathcal{M} \subset \mathbb{R}^{D_1 \times \cdots \times D_k}$. In

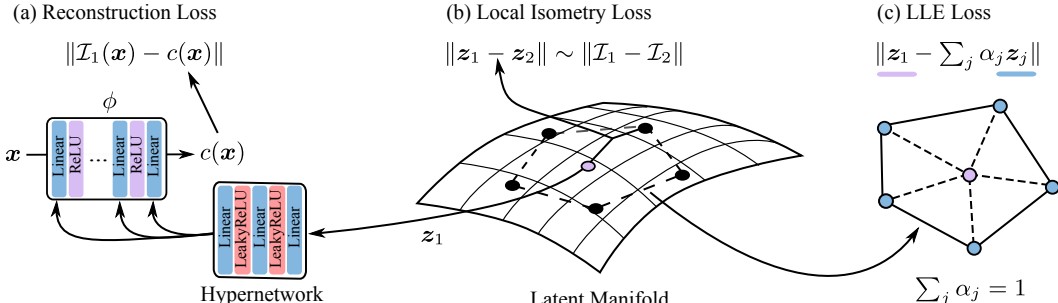

Figure 2: GEM learns to embed an arbitrary signal $\mathcal{I}$ in a latent manifold via three losses: (a) a reconstruction loss which encourages neural network $\phi$, decoded from $z_1$ through a hypernetwork, to match training signal $\mathcal{I}_1$ at each coordinate $\boldsymbol{x}$, (b) a local isometry loss to enforce perceptual consistency, which encourages distance between nearby latents $z_1$ and $z_2$ to be porportional to their distance in signal space $\mathcal{I}_1$, $\mathcal{I}_2$, (c) a LLE loss which encourages a latent $z_1$ to be represented as a convex combination $\alpha_i$ of its neighbors.

particular, a manifold $\mathcal{M}$ [2, 3], over signals $\mathcal{I}$, is a locally low-dimensional subspace consisting and describing all possible variations of $\mathcal{I}$. A manifold has three crucial properties. First, it must embed all possible instances of $\mathcal{I}$. Second, $\mathcal{M}$ should be locally metric around each signal $\mathcal{I}$, enabling us to effectively navigate and interpolate between nearby signals. Finally, it must be locally perceptually consistent; nearby points on $\mathcal{M}$ should be similar in identity. To learn a manifold $\mathcal{M}$ with these properties, we introduce a loss for each property: $\mathcal{L}_{\text{Rec}}$, $\mathcal{L}_{\text{LLE}}$, and $\mathcal{L}_{\text{Iso}}$, respectively. $\mathcal{L}_{\text{Rec}}$ ensures that all signals $\mathcal{I}$ are embedded inside $\mathcal{M}$ (Section 4.1), $\mathcal{L}_{\text{LLE}}$ forces the space near each $\mathcal{I}$ to be locally metric (Section 4.2), and $\mathcal{L}_{\text{Iso}}$ encourages the underlying manifold to be perceptually consistent (Section 4.3). As such, our training loss takes the form:

$$\mathcal{L}_{\text{Total}} = \mathcal{L}_{\text{Rec}} + \mathcal{L}_{\text{LLE}} + \mathcal{L}_{\text{Iso}} \tag{3}$$

Please see Fig. 2 for an overview of the proposed manifold learning approach. We utilize equal weighting across each introduced loss term. We discuss each component of the loss in more detail below.

## 4.1 Data Coverage via Auto-Decoding

Given a parameterization of a subspace $\mathcal{F}$ (as introduced in Section 3) which represents our manifold $\mathcal{M}$, we must ensure that $\Psi$, our hypernetwork, can effectively cover the entire subspace in which our signal $\mathcal{I}$ lies. A common approach to learn subspaces over signals is to employ a GAN [4], whereby a discriminator trains a hypernetwork $\Psi(z)$ to generate signals that are indistinguishable from real examples. However, an issue with such an approach is that many $\mathcal{I}$ may be missing in the resultant mapping.

Here, to ensure that our manifold covers all instances, we utilize the auto-decoder framework [17, 21]. We learn a separate latent $z_i$ for each training signal $\mathcal{I}_i$ and train models utilizing the loss

$$\mathcal{L}_{\text{Rec}} = \|\Psi(z_i)(\boldsymbol{x}) - \mathcal{I}_i(\boldsymbol{x})\|^2 \tag{4}$$

over each possible input coordinate $\boldsymbol{x}$. By explicitly learning to reconstruct each training point in our latent space, we enforce that our latent space, and thus our manifold $\mathcal{M}$, covers all training signals $\mathcal{I}$.

## 4.2 Local Metric Consistency via Linear Decomposition

A manifold $\mathcal{M}$ is locally metric around a signal $\mathcal{I}$ if there exist local linear directions of variation on the manifold along which the underlying signals vary in a perceptually coherent way. Inspired by [3], we enforce this constraint by encouraging our manifold to consist of a set of local convex linear regions $\mathcal{R}_i$ in our learned latent space.

We construct these convex linear regions $\mathcal{R}_i$ using autodecoded latents $z_i$ (Section 4.1). Each latent $z_i$ defines a convex region, $\mathcal{R}_i$ – obtained by combining nearest latent neighbors $z_i^1, \ldots, z_i^j$, as illustrated in Figure 2.

To ensure that our manifold $\mathcal{M}$ consists of these convex local regions, during training, we enforce that each latent $z_i$ can be represented in a local convex region, $\mathcal{R}_i$, e.g. that it can be expressed as a

set of weights $\boldsymbol{w}_i$, such that $\boldsymbol{z}_i = \sum_j w_i^j \boldsymbol{z}_i^j$ and $\sum_j w_i^j = 1$. Given a $\boldsymbol{z}_i$ we may solve for a given $\boldsymbol{w}_i$ by minimizing the objective $\|\boldsymbol{z}_i - \sum_j w_i^j \boldsymbol{z}_i^j\|^2 = \boldsymbol{w}_i^T \boldsymbol{G}_i \boldsymbol{w}_i$ with respect to $\boldsymbol{w}_i$. In the above expression, $\boldsymbol{G}_i$ is the Gram matrix, and is defined as $\boldsymbol{G}_i = \boldsymbol{k}\boldsymbol{k}^T$, where $\boldsymbol{k}$ is a block matrix, with row $j$ corresponding to $\boldsymbol{k}_j = \boldsymbol{z}_i - \boldsymbol{z}_i^j$. Adding the constraint that all weights add up to one via a Legrange multiplier, leads to the following optimization objective:

$$\mathcal{L}(\boldsymbol{w}_i, \lambda) = \boldsymbol{w}_i^T \boldsymbol{G}_i \boldsymbol{w}_i + \lambda(\boldsymbol{1}^T \boldsymbol{w}_i - 1) \tag{5}$$

where $\boldsymbol{1}$ corresponds to a vector of all ones. Finding the optimal $\boldsymbol{w}_i$ in the above expression then corresponds to the best projection of $\boldsymbol{z}_i$ on the linear region $\mathcal{R}_i$ it defines. Taking gradients of the above expression, we find that $\boldsymbol{w}_i = \frac{\lambda}{2} \boldsymbol{G}_i^{-1} \boldsymbol{1}$, where $\lambda$ is set to ensure that elements of $\boldsymbol{w}_i$ sum up to 1. The mapped latent $\boldsymbol{z}_i' = \sum_j \boldsymbol{w}_i^j \boldsymbol{z}_i^j$ corresponds to the best projection $\boldsymbol{z}_i$ on $\mathcal{R}_i$. To enforce that $\boldsymbol{z}_i$ lies in $\mathcal{R}_i$, we enforce that this projection

$$\mathcal{L}_{\text{LLE}} = \|\Psi(\boldsymbol{z}_i')(\boldsymbol{x}) - \mathcal{I}_i(\boldsymbol{x})\|^2, \tag{6}$$

also decodes to the training signal $\mathcal{I}_i$, where we differentiate through intermediate matrix operations. To encourage $\boldsymbol{w}_i$ to be positive, we incorporate an additional L1 penalty to negative weights.

### 4.3 Perceptual Consistency through Local Isometry

Finally, a manifold $\mathcal{M}$ should be perceptually consistent. In other words, two signals that are perceptually similar should also be closer in the underlying manifold, according to a distance metric on the respective latents, than signals that are perceptually distinct. While in general the MSE distance between two signals is not informative, when such signals are relatively similar to one another, their MSE distance *does* contain useful information.

Therefore, we enforce that the distance between latents $\boldsymbol{z}_i$ and $\boldsymbol{z}_j$ in our manifold $\mathcal{M}$ are locally isometric to the MSE distance of the corresponding samples $\mathcal{I}_i$ and $\mathcal{I}_j$, when relative pairwise distance of samples is small. On the lowest 25% of pairwise distances, we utilize the following loss:

$$\mathcal{L}_{\text{Iso}} = \|\alpha * \|(\boldsymbol{z}_i - \boldsymbol{z}_j)\| - \|\mathcal{I}_i - \mathcal{I}_j\|\|. \tag{7}$$

Such local isometry may alternatively be enforced by regularizing the underlying Jacobian map.

### 4.4 Applications

By learning a modality-independent manifold $\mathcal{M}$, GEM permits the following *signal-agnostic* downstream applications, each of which we explore in this work.

**Interpolation.** By directly interpolating between the learned latents $\boldsymbol{z}_i$ and $\boldsymbol{z}_j$ representing each signal, we can interpolate in a perceptually consistent manner between neighboring signals in $\mathcal{M}$. Further, latents for novel signals may be obtained by optimizing $\mathcal{L}_{\text{Rec}}$ on the signal.

**Conditional Completion.** Given a partial signal $\hat{\mathcal{I}}$, we can obtain a point on the manifold $\mathcal{M}$ by optimizing a latent $\boldsymbol{z}$ using $\mathcal{L}_{\text{Rec}} + \mathcal{L}_{\text{LLE}}$. Decoding the corresponding latent $\boldsymbol{z}$ then allows us to complete the missing portions of the image.

**Sample Generation.** Our manifold $\mathcal{M}$ consists of a set of locally linear regions $\mathcal{R}_i$ in a underlying latent space $\boldsymbol{z}$. As such, new samples in $\mathcal{M}$ can be generated by walking along linear regions $\mathcal{R}_i$, and sampling points each such region. To sample a random point within $\mathcal{R}_i$, we generate $\alpha$ between 0 and 1, and compute a latent $\hat{\boldsymbol{z}}_i = \alpha \boldsymbol{z}_i + (1 - \alpha) * \boldsymbol{z}_i^j + \beta * \mathcal{N}(0, 1)$ (corresponding to a random sample in the neighborhood of $\boldsymbol{z}_i$) and project it onto the underlying region $\mathcal{R}_i$.

## 5 Learning Manifolds of Data

We validate the generality of GEM by first showing that our model is capable of fitting diverse signal modalities. Next, we demonstrate that our approach captures the underlying structure across these signals; we are not only able to cluster and perceptually interpolate between signals, but inpaint to complete partial ones. Finally, we show that we can draw samples from the learned manifold of each signal type, illustrating the power of GEM to be used a signal-agnostic generative model. We re-iterate that *nearly identical* architectures and training losses are used across separate modalities.

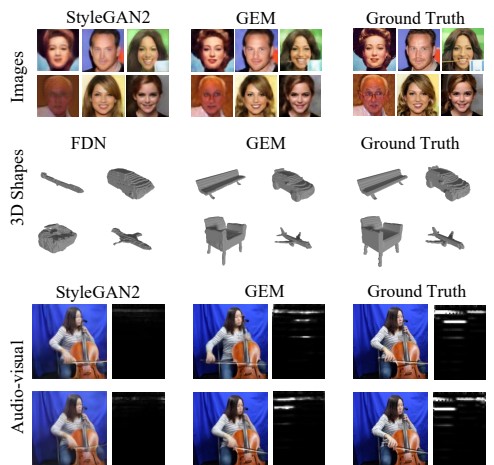

| Modality | Model | MSE ↓ | PSNR ↑ |
|---|---|---|---|
| Images | Heat Kernel | 0.0365 | 14.99 |
| | VAE | 0.0331 | 15.12 |
| | FDN | 0.0060 | 22.59 |
| | StyleGAN2 | 0.0044 | 24.03 |
| | GEM | **0.0025** | **26.53** |
| Audio | VAE | 0.0147 | 18.87 |
| | FDN | 0.0050 | 23.68 |
| | StyleGAN2 | 0.0015 | 28.52 |
| | GEM | **0.0011** | **29.98** |
| ShapeNet | FDN | 0.0296 | 16.52 |
| | GEM | **0.0153** | **21.32** |
| Image and Audio | VAE | 0.0193 | 17.23 |
| | FDN | 0.0663 | 11.78 |
| | StyleGAN2 | 0.0063 | 22.36 |
| | GEM | **0.0034** | **24.38** |

Figure 3: **Test-time reconstruction.** Comparison of GEM against baselines on the task of fitting test signals over a diverse set of signal modalities: images, 3D shapes, and audio-visual, respectively. GEM (center) achieves significantly better reconstruction performance – qualitatively and quantitatively – on test signals than StyleGAN2 or FDN, indicating better coverage of the data manifold across all modalities. Results are run across one seed, but we report results across three seeds in the appendix, and find limited overall variance.

**Datasets.** We evaluate GEM on four signal modalities: image, audio, 3D shape, and cross-modal image and audio signals, respectively. For the image modality, we investigate performance on the CelebA-HQ dataset [38] fit on 29000 $64 \times 64$ training celebrity images, and test on 1000 $64 \times 64$ test images. To study GEM behavior on audio signals, we use the NSynth dataset [39], and fit on a training set of 10000 one-second 16kHz sounds clips of different instruments playing, and test of 5000 one-second 16kHz sound clips. We process sound clips into spectrograms following [32]. For the 3D shape domain, we work with the ShapeNet dataset from [40]. We train on 35019 training shapes at $64 \times 64 \times 64$ resolution and test on 8762 shapes at $64 \times 64 \times 64$ resolution. Finally, for the cross-modal image and audio modality, we utilize the cello image and audio recordings from the Sub-URMP dataset [41]. We train on 9800 images at $128 \times 128$ resolution and 0.5 second 16kHz audio clips, which are also processed into spectrograms following [32] and test on 1080 images and associated audio clips. We provide additional cross-modal results in the supplement.

**Setup and Baselines.** We benchmark GEM against existing manifold learning approaches. Namely, we compare our approach with that of StyleGAN2 [31], VAE [42], and concurrent work FDN of [30], as well as heat kernel manifold learning [14]. We utilize the authors' provided codebase for StyleGAN2, the PytorchVAE library [3] for the VAE, and the original codebase of FDN [30], which the authors graciously provided. We train all approaches with a latent dimension of 1024, and re-scale the size of the VAE to ensure parameter counts are similar. We report model architecture details in the appendix. We need to change the architecture of both the VAE and StyleGAN2 baselines, respectively, in order to preserve proper output dimensions, depending on signal modality. However, we note that identical architectures readily can and are used for GEM. For fitting cross-modal audiovisual signals, all models utilize two separate decoders, with each decoder in GEM identical to each other. Architectural details are provided in the appendix.

## 5.1 Covering the Data Manifold

We first address whether our model is capable of covering the underlying data distribution of each modality, compared against our baselines, and study the impact of our losses on this ability.

**Modality Fitting.** We quantitatively measure reconstruction performance on test signals in Figure 3 and find the GEM outperforms each of our baselines. In the cross-modal domain in particular, we find that FDN training destabilizes quickly and struggles to adequately fit examples. We illustrate test signal reconstructions in Figure 3 and Figure 4. Across each modality, we find that GEM obtains the sharpest samples. In Figure 3 we find that while StyleGAN2 can reconstruct certain CelebA-HQ

---

[3]https://github.com/AntixK/PyTorch-VAE

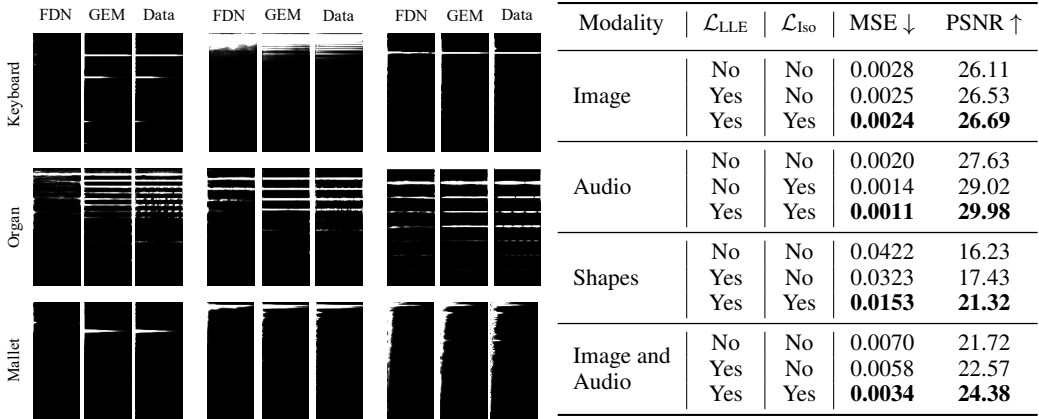

| Modality | $\mathcal{L}_{\text{LLE}}$ | $\mathcal{L}_{\text{Iso}}$ | MSE ↓ | PSNR ↑ |
|---|---|---|---|---|
| Image | No | No | 0.0028 | 26.11 |
|  | Yes | No | 0.0025 | 26.53 |
|  | Yes | Yes | **0.0024** | **26.69** |
| Audio | No | No | 0.0020 | 27.63 |
|  | No | Yes | 0.0014 | 29.02 |
|  | Yes | Yes | **0.0011** | **29.98** |
| Shapes | No | No | 0.0422 | 16.23 |
|  | Yes | No | 0.0323 | 17.43 |
|  | Yes | Yes | **0.0153** | **21.32** |
| Image and Audio | No | No | 0.0070 | 21.72 |
|  | Yes | No | 0.0058 | 22.57 |
|  | Yes | Yes | **0.0034** | **24.38** |

Figure 4: **Test-time audio reconstruction.** GEM (center) achieves better spectrogram recovery than FDN (left), even recovering fine details in the original signals.

Table 1: **Ablation.** Impact of $\mathcal{L}_{\text{Iso}}$ and $\mathcal{L}_{\text{LLE}}$ on test signal reconstruction performance. Ablating each loss demonstrates that both components enable better fitting of the underlying test distribution.

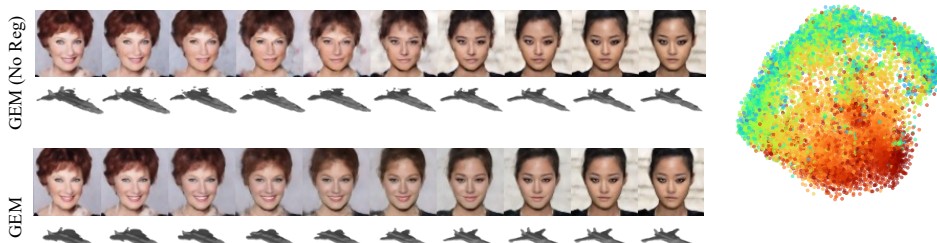

Figure 5: **Nearest Neighbor Interpolation.** Interpolation between nearest neighbors in GEM, in the images and 3D shape domains. Our model is able to make perceptually continuous interpolations.

Figure 6: **t-SNE Structure.** t-SNE plot of NSynth audio clip manifold embeddings colored by corresponding pitch. Pitch is seperated by t-SNE.

images well, it fails badly on others, indicating a lack of data coverage. Our results indicate that our approach best captures the underlying data distribution.

**Ablations.** In Table 1, we assess the impact of two of our proposed losses – $\mathcal{L}_{\text{LLE}}$ and $\mathcal{L}_{\text{Iso}}$ – at enabling test-signal reconstruction, and find to both help improve the resultant test construction. We posit that both losses serve to regularize the underlying learned data manifold, enabling GEM to more effectively cover the overall signal manifold.

**Applications to Other Models.** While we apply $\mathcal{L}_{\text{Rec}}$, $\mathcal{L}_{\text{LLE}}$, $\mathcal{L}_{\text{Iso}}$ to GEM, our overall losses may be applied more generally across different models, provided the model maps an underlying latent space to an output domain. We further assess whether these losses improve the manifold captured by StyleGAN2 via measuring reconstruction performance on test CelebA-HQ images. We find that while a base StyleGAN2 model obtains a reconstruction error of MSE 0.0044 and PSNR 24.03, the addition of $\mathcal{L}_{\text{Rec}}$ improves test reconstruction to MSE 0.0041 and PSNR 24.29 with $\mathcal{L}_{\text{LLE}}$ losses further improving test reconstruction to MSE 0.0038 and PSNR 24.61. In contrast, we found that $\mathcal{L}_{\text{Iso}}$ leads to a slight regression in performance. We posit that both $\mathcal{L}_{\text{Rec}}$ and $\mathcal{L}_{\text{LLE}}$ serve to structure the underlying StyleGAN2 latent space, while the underlying Gaussian nature of the StyleGAN2 latent space precludes the need for $\mathcal{L}_{\text{Iso}}$. Our results indicate the generality of our proposed losses towards improving the recovery of data manifolds.

## 5.2 Learning the Structure of Signals

Next, we explore the extent to which the manifold learned by GEM captures the underlying global *and* local structure inherent to each signal modality. Additionally, we probe our model's understanding of individual signals by studying GEM's ability to reconstruct partial signals.

**Global Structure.** We address whether GEM captures the global structure of a signal distribution by visualizing the latents learned by GEM on the NSynth dataset with t-SNE [43]. We find in Figure 6

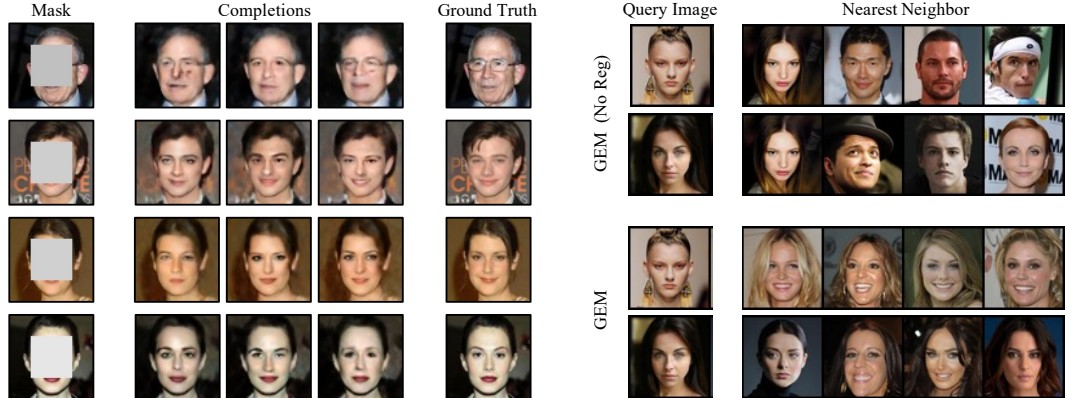

Figure 7: **Diverse Image Inpainting (Left).** GEM generates multiple possible completions of a partial image. Across completions, skin and lip hues are completed correctly and eyeglasses are reconstructed consistently. **Nearest Neighbors (Right).** Nearest neighbors in the manifold of GEM with and without regularization. With regularization, neighbors in latent space correspond to perceptually similar images.

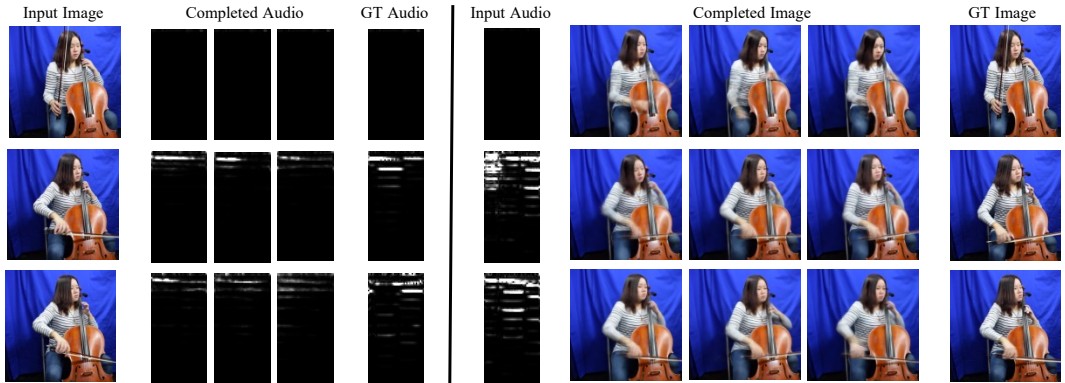

Figure 8: **Audio Hallucinations (Left).** GEM generates multiple possible audio spectrogram completions of an input image. Generations are cross-modal. For instance, conditioned on an image of a lifted bow, empty spectrograms are generated; likewise, higher frequency spectrograms are generated when the bow is on high strings, and lower frequency ones on lower strings. **Image Hallucinations (Right).** Given an input spectrogram, GEM generates multiple possible corresponding image completions, again highlighting the cross-modality of the manifold. When the input spectrogram is empty, GEM generates images having the bow off the cello, and depending on the frequency composition of the spectrogram, the bow position is sampled along different strings.

that the underlying pitch of a signal maps onto the resulting t-SNE. We further visualize the inferred connectivity structure of our manifold in Figure 7 (left) and visualize the nearest neighbors in latent space of the autodecoded latents (right), finding that nearby latents correspond to semantically similar faces – further supporting that GEM learns global structure. Note, we find that without either $\mathcal{L}_{\text{LLE}}$ or $\mathcal{L}_{\text{Iso}}$, the underlying connectivity of our manifold is significantly poorer.

**Local Structure.** We next probe whether GEM learns a densely connected manifold, e.g. one which allows us to interpolate between separate signals, suggestive of capturing the local manifold structure. In Figure 5, we visualize nearest neighbor interpolations in our manifold and observe that the combination of $\mathcal{L}_{\text{LLE}}$ and $\mathcal{L}_{\text{Iso}}$ enables us to obtain perceptually smooth interpolations over both image and shape samples. We provide additional interpolations and failure cases in the supplement.

**Signal Completion.** To investigate the ability of GEM to understand individual signals, we assess the recovery of a full signal when only a subset of such a signal is given. In Figure 7, we consider the case of inpainting CelebA-HQ faces, where a subset of the face is missing. We observe that GEM is able to obtain several different perceptually consistent inpaintings of a face, with individual completions exhibiting consistent skin and lip colors, and restoring structural features such as eyeglasses. We provide additional examples and failure cases in the supplement.

Additionally, we consider signal completion in the cross-modal audiovisual domain. Here, we provide only an input image and ask GEM to generate possible audio spectrograms. As seen in Figure 8, we

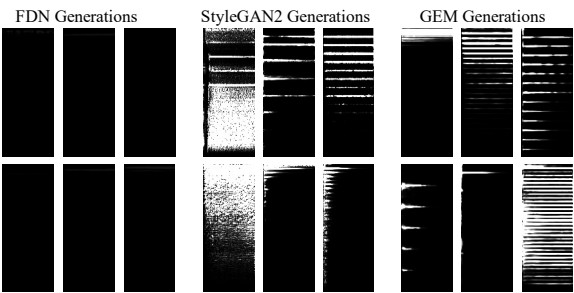

| Model | FID ↓ | Precision ↑ | Recall ↑ |
|---|---|---|---|
| **CelebA-HQ 64x64** | | | |
| VAE | 175.33 | **0.799** | 0.001 |
| StyleGAN V2 | **5.90** | 0.618 | 0.481 |
| FDN [30] | 13.46 | 0.577 | 0.397 |
| GEM | 30.42 | 0.642 | **0.502** |
| **ShapeNet** | Coverage ↑ | MMD ↓ | |
| Latent GAN [40] | 0.389 | 0.0017 | |
| FDN [30] | 0.341 | 0.0021 | |
| GEM (Ours) | **0.409** | **0.0014** | |

Table 2: **Generation Performance.** Performance of GEM and baselines on signal generation.

Figure 9: **Audio Generations.** GEM (right) generates sharp audio samples compared to FDN and StyleGAN2.

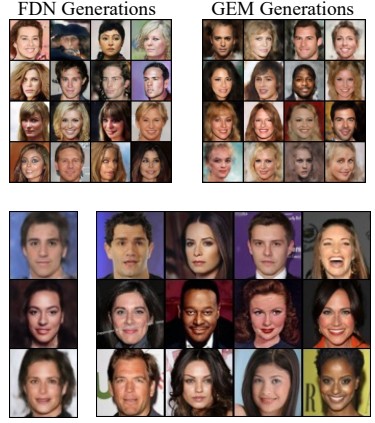

Figure 10: **Image Generations (Top).** GEM (right) generates comparable images to FDN. **Nearest Neighbors (Bottom).** Illustration of nearest neighbors in latent space of generations.

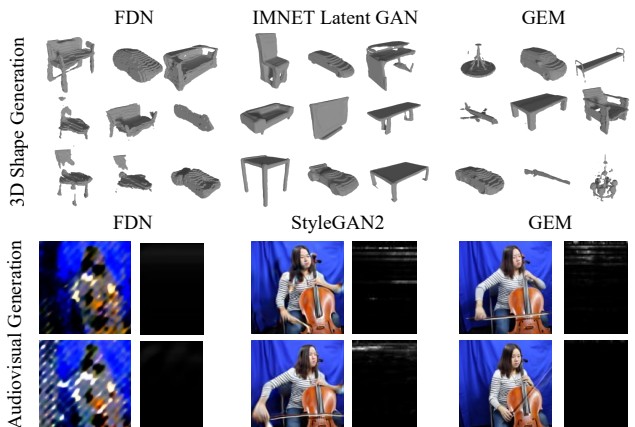

Figure 11: **Generations.** GEM (right) produces reasonable samples across the shape and audiovisual modalities. In contrast, audiovisual generations from StyleGAN2 exhibit noise, while FDN generates poor samples in both modalities.

observe that GEM is able to generate empty spectograms when the bow is not on the cello, as well as different spectrograms dependent on the position of the bow. Alternatively, in Figure 8, we provide only an input audio spectrogram and ask GEM to generate possible images. In this setting, we find that generated images have the bow off the cello when the input spectrogram is empty (with the bow missing due to GEM blurring the uncertainty of all possible locations of the bow), and at different positions depending on the input audio spectrogram. In contrast, we found that all baselines were unable to fit the audiovisual setting well, as discussed in Section 5.1 and shown in Figure 11.

## 5.3 Generating Data Samples

Finally, we investigate the ability of GEM to generate new data samples.

**Qualitative Generations.** We show random samples drawn from GEM and compare these to our baselines. We consider images in Figure 10, shapes in Figure 11, audio snippets in Figure 9 and audiovisual inputs in Figure 11. While we find that GEM performs comparably to FDN in the image regime, our model significantly outperforms all baselines on domains of audio, shape, and audiovisual modalities. We further display nearest latent space neighbors of generations on CelebA-HQ in Figure 10 and find that our generations are distinct from those in the training dataset.

**Quantitative Comparisons.** Next, we provide quantitiative evaluations of generations in Table 2 on image and shape modalities. We report the FID [44], precision, and recall [45] metrics on CelebA-HQ $64 \times 64$. We find that GEM performs better than StyleGAN2 and FDN on precision and recall, but worse in terms of FID. We note that our qualitative generations in Figure 10 are comparable to those of FDN; however, we find that our approach obtains high FID scores due to the inherent sensitivity of the FID metric to blur. Such sensitivity to bluriness has also been noted in [46], and we

find that even our autodecoded training distribution obtains an FID of 23.25, despite images appearing near perceptually perfect (Figure 3). On ShapeNet, we report coverage and MMD metrics from [47] using Chamfer Distance. To evaluate generations, we sample 8762 shapes (the size of test Shapenet dataset) and generate 2048 points following [40]. We compare generations from GEM with those of FDN and latent GAN trained on IMNET (using the provided code in [40]). We find that GEM outperforms both approaches on the task of 3D shape generation.

## 6 Conclusion

We have presented GEM, an approach to learn a modality-independent manifold. We demonstrate how our model enables us to interpolate between signals, complete partial signals, and further generate new signals. A limitation of our approach is that while underlying manifolds are recovered, they are not captured with high-fidelity. We believe that a promising direction of future work involves the pursuit of further engineered inductive biases, and general structure, which enable the recovery of high-fidelity manifolds of the plethora of signals in the world around us. We emphasize, however, that as our manifold is constructed over a dataset; GEM may learn to incorporate and propogate existing prejudices and biases present in such data, posing risks for mass deployment of our model. Additionally, while our work offers exciting avenues for cross-modal modeling and generation, we note that GEM has the potential to be used to create enhanced "deep fakes" and other forms of synthetic media.

**Acknowledgements**   This project is supported by DARPA under CW3031624 (Transfer, Augmentation and Automatic Learning with Less Labels), the Singapore DSTA under DST00OECI20300823 (New Representations for Vision), as well as ONR MURI under N00014-18-1-2846. We would like to thank Bill Freeman and Fredo Durand for giving helpful comments on the manuscript. Yilun Du is supported by an NSF graduate research fellowship.

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
