# A  Appendix for Learning Signal-Agnostic Manifolds of Neural Fields

Please visit our project website at https://yilundu.github.io/gem/ for additional qualitative visualizations of test-time reconstruction of audio and audiovisual samples, traversals along the underlying manifold of GEM on CelebA-HQ as well as interpolations between audio samples. We further illustrate additional image in-painting results, as well as audio completion results. Finally, we visualize several audio and audiovisual generations.

In Section A.1 below, we provide details on training settings, as well as the underlying baseline model architectures utilized for each modality. We conclude with details on reproducing our work in Section A.2.

## A.1  Experimental Details

**Training Details**    For each separate training modality, all models and baselines are trained for one day, using one 32GB Volta machine. GEM is trained with the Adam optimizer [3], using a training batch size of 128 and a learning rate of 1e-4. Each individual datapoint is fit by fitting the value of 1024 sampled points in the sample (1024 for each modality in the multi-modal setting). We normalize the values of a signals to be between -1 and 1. When computing $\mathcal{L}_{\text{Iso}}$, a scalar constant of $\alpha = 100$ is employed to scale distances in the underlying manifold to that of distances of signals in sample space. When enforcing $\mathcal{L}_{\text{LLE}}$, a total of 10 neighbors are considered to compute the loss across modalities. We utilize equal loss weight across $\mathcal{L}_{\text{Rec}}$, $\mathcal{L}_{\text{Iso}}$, $\mathcal{L}_{\text{LLE}}$, and found that the relative magnitudes of each loss had little impact on the overall performance.

**Model Details**    We provide the architectures of the hypernetwork $\psi$ and implicit function $\phi$ utilized by GEM across separate modalities in Table 2 and Table 3, respectively. Additionally, we provide the architectures used in each domain for our baselines: StyleGAN2 in Table 13 each domain in Table 13 and VAE in Table 8. Note that for the VAE, the ReLU nonlinearity is used, with each separate convolution having stride 2.

We obtained the hyperparameters for implicit functions and hypernetworks based off of [5]. Early in the implementation of the project, we explored a variety of additional architectural choices; however, we ultimately found that neither changing the number of layers in the hypernetworks, nor changing the number of underlying hidden units in networks, significantly impacted the performance of GEM. We will add these details to the appendix of the paper.

## A.2  Reproducibility

We next describe details necessary to reproduce each of other underlying empirical results.

**Hyperparameter Settings for Baselines**    We employ the default hyperparameters, as used in the original papers for StyleGAN2 [2] and FDN [1], to obtain state-of-the-art performance on their respective tasks. Due to computational constraints, we were unfortunately unable to do a complete hyperparameter search for each method over all tasks considered. Despite this, we were able to run the models on toy datasets and found that these default hyperparameters performed the best. We utilized the author's original codebases for experiments.

**Variance Across Seeds**    Results in the main tables of the paper are run across a single evaluated seed. Below in Table 1, we rerun test reconstruction results on CelebA-HQ across different models utilizing a total of 3 separate seeds. We find minimal variance across separate runs, and still find the GEM performs significantly outperforms baselines.

| Modality | Model | MSE ↓ | PSNR ↑ |
|---|---|---|---|
| Images | VAE | $0.0327 \pm 0.0035$ | $15.16 \pm 0.06$ |
| | FDN | $0.0062 \pm 0.0003$ | $22.57 \pm 0.02$ |
| | StyleGANv2 | $0.0044 \pm 0.0001$ | $24.03 \pm 0.01$ |
| | GEM | $\mathbf{0.0025} \pm 0.0001$ | $\mathbf{26.53} \pm 0.01$ |

Table 1: Test CelebA-HQ reconstruction results of different methods evaluated across 3 different seeds. We further report standard deviation between different runs.

**Datasets** We provide source locations to download each of the datasets we used in the paper. The CelebA-HQ dataset can be downloaded at https://github.com/tkarras/progressive_growing_of_gans/blob/master/dataset_tool.py and is released under the Creative Commons license. The NSynth dataset may be downloaded at https://magenta.tensorflow.org/datasets/nsynth and is released under the Creative Commons license. The ShapeNet dataset can be downloaded at https://github.com/czq142857/IM-NET and is released under the MIT License, and finally the Sub-URMP dataset we used may be downloaded at https://www.cs.rochester.edu/~cxu22/d/vagan/.

| Dense → 512 |
| --- |
| Dense → 512 |
| Dense → 512 |
| Dense → $\phi$ Parameters |

Table 2: The architecture of the hypernetwork utilized by GEM.

| Pos Embed (512) |
| --- |
| Dense → 512 |
| Dense → 512 |
| Dense → 512 |
| Dense → Output Dim |

Table 3: The architecture of the implicit function $\phi$ used to agnostically encode each modality. We utilize the Fourier embedding from [4] to embed coordinates.

| 3x3 Conv2d, 64 |
| --- |
| 3x3 Conv2d, 128 |
| 3x3 Conv2d, 256 |
| 3x3 Conv2d, 512 |
| 3x3 Conv2d, 512 |
| $z \leftarrow$ Encode |
| Reshape(2, 2) |
| 3x3 Conv2d Transpose, 512 |
| 3x3 Conv2d Transpose, 512 |
| 3x3 Conv2d Transpose, 256 |
| 3x3 Conv2d Transpose, 128 |
| 3x3 Conv2d Transpose, 3 |

Table 4: The encoder and decoder of the VAE utilized for CelebA-HQ.

| 3x3 Conv2d, 32 |
| --- |
| 3x3 Conv2d, 64 |
| 3x3 Conv2d, 128 |
| 3x3 Conv2d, 256 |
| 3x3 Conv2d, 512 |
| $z \leftarrow$ Encode |
| Reshape(4, 2) |
| 3x3 Conv2d Transpose, 512 |
| 3x3 Conv2d Transpose, 256 |
| 3x3 Conv2d Transpose, 128 |
| 3x3 Conv2d Transpose, 64 |
| 3x3 Conv2d Transpose, 32 |
| 3x3 Conv2d Transpose, 1 |
| Crop |

Table 5: The encoder and decoder of the VAE utilized for NSynth

| 3x3 Conv2d, 32 |
| --- |
| 3x3 Conv2d, 64 |
| 3x3 Conv2d, 128 |
| 3x3 Conv2d, 256 |
| 3x3 Conv2d, 512 |
| $z \leftarrow$ Encode |
| Reshape(2, 2) |
| 3x3 Conv2d Transpose, 512 |
| 3x3 Conv2d Transpose, 256 |
| 3x3 Conv2d Transpose, 128 |
| 3x3 Conv2d Transpose, 64 |
| 3x3 Conv2d Transpose, 32 |
| 3x3 Conv2d Transpose, 3 |

Table 6: The architecture of encoder and decoder of the VAE utilized for audiovisual dataset on images. Latent encodings from image and audio modalities are added together.

| 3x3 Conv2d, 32 |
| --- |
| 3x3 Conv2d, 64 |
| 3x3 Conv2d, 128 |
| 3x3 Conv2d, 256 |
| 3x3 Conv2d, 512 |
| $z \leftarrow$ Encode |
| Reshape(4, 1) |
| 3x3 Conv2d Transpose, 512 |
| 3x3 Conv2d Transpose, 256 |
| 3x3 Conv2d Transpose, 128 |
| 3x3 Conv2d Transpose, 64 |
| 3x3 Conv2d Transpose, 32 |
| 3x3 Conv2d Transpose, 1 |
| Crop |

Table 7: The architecture of encoder and decoder of the VAE utilized for audiovisual dataset on audio. Latent encodings from image and audio modalities are added together.

Table 8: The architecture of the VAE utilized across datasets.

| Constant Input (512, 4, 4) |
| --- |
| StyleConv 512 |
| StyleConv 512 |
| StyleConv 512 |
| StyleConv 512 |
| StyleConv 256 |
| 3x3 Conv2d, 3 |

Table 9: The generator architecture of Style-GAN2 for CelebA-HQ.

| Constant Input (512, 8, 4) |
| --- |
| StyleConv 512 |
| StyleConv 512 |
| StyleConv 512 |
| StyleConv 512 |
| StyleConv 256 |
| 3x3 Conv2d, 1 |
| Crop |

Table 10: The generator architecture of Style-GAN2 for NSynth

| Constant Input (512, 4, 4) |
| --- |
| StyleConv 512 |
| StyleConv 512 |
| StyleConv 512 |
| StyleConv 512 |
| StyleConv 256 |
| 3x3 Conv2d, 3 |

Table 11: The generator architecture of Style-GAN2 for audiovisual domain for images.

| Constant Input (512, 8, 2) |
| --- |
| StyleConv 512 |
| StyleConv 512 |
| StyleConv 512 |
| StyleConv 512 |
| StyleConv 256 |
| 3x3 Conv2d, 1 |
| Crop |

Table 12: The generator architecture of Style-GAN2 for audiovisual domain for audio.

Table 13: The architecture of the StyleGAN generator utilized across datasets.