# OpenReview forum: "Learning Signal-Agnostic Manifolds of Neural Fields"
_NeurIPS.cc/2021/Conference — NeurIPS 2021 Poster_

### Official Review · Reviewer_oxSK · 2021-07-14

**Rating:** 6
**Confidence:** 4

**Summary:**

This work introduces GEM - a model which learns the low-dimensional latent manifold structure of a dataset in a modality-independent way. GEM’s objective function is designed to encourage the learning of manifolds which exhibit the following properties, namely they: cover the data, are locally metric, and are globally consistent.  The model is trained on four datasets with different modalities (image, audio, 3D shape, and cross-modal image and audio). Qualitative and quantitative evaluations suggest that GEM can be used to do linear interpolation between points on the learned manifold, it can generate reconstructions of corrupted samples as well as generate new samples.

**Limitations And Societal Impact:**

Yes, they have.

**Main Review:**

### Strengths

- To the best of my knowledge, this is a novel approach to manifold learning which can easily be applied to data from different modalities.
- The evaluation protocol is extensive and is both qualitative and quantitative. It includes visualizations of interpolation between signals, signal generation, and signal inpainting (for images), as well as hallucinations of one modality given another.
- The paper is organized and written clearly.

### Room for Improvement and Questions
1. The presented quantitative results are missing error bars with respect to the number of random seeds used even though the answer to this question on the Checklist is [YES].
2. How were the hidden layer size (of 512) and number of layers in the implicit function $\Phi$ and hypernetwork $\Psi$ chosen? Have the authors tried other values for these hyperparameters and measured their effect on the reported results?
3.  How was hyperparameter search performed for the benchmarks (StyleGAN2, FDN, ShapeNet)?
4. Since GEM’s loss is a sum of three different components, how are the relative weights of each loss component determined?
5. I noticed that in the Image Hallucination visualization in Fig 8 for an empty spectrogram the bow is very blurry or missing. Do the authors have a guess of what the reason for this might be?

### Nitpicks
- Line 72: “embeddings $\it{obtained}$”
- Figure 1: “represents $\it{an}$ audio spectrogram”
- Line 102: there is an extra “method”
- Line 106: “coordinate”
- Line 111: “the same across signal types” (without “for”)
- Line 120: $W^{L_{h}}$ should be $W^{L}_{h}$
- Line 129: there is an extra “in”
- Line 172: “than signals that $\it{are}$ perceptually distinct”

### Rebuttal Period Update
I've decided to increase my score to 6 after reading the author's response to my questions and concerns.

**Time Spent Reviewing:**

5

---

> ### Author Response · Authors · 2021-08-10
> **Reviewer oxSK Response**
>
> Thank you for your detailed comments and feedback; we appreciate the time and attention you gave to reviewing our paper. We resolve concerns about experimental settings, hyperparameters, and results below, and we will update the paper following each of your suggested comments.  Please let us know if you have any additional questions, we are happy to clarify or provide additional experiments.
>
> ------------------------
>
> **Q1) Confidence Errors of Quantitative Results.**
> Please see Author Response, Section 2.5.
>
> **Q2) Size of Implicit Functions and Hypernetworks.**
> We obtained the hyperparameters for implicit functions and hypernetworks based off of [36] . Early in the implementation of the project, we explored a variety of additional architectural choices; however, we ultimately found that neither changing the number of layers in the hypernetworks, nor changing the number of underlying hidden units in networks, significantly impacted the performance of GEM. We will add these details to the appendix of the paper.
>
> **Q3) Baseline Hyperparameters.**
>  We employ the default hyperparameters, as used in the original papers for StyleGAN2 [30] and  FDN [29], to obtain state-of-the-art performance on their respective tasks. Due to computational constraints, we were unfortunately unable to do a complete hyperparameter search for each method over all tasks considered. Despite this, we were able to run the models on toy datasets and found that these default hyperparameters performed the best.  We note that we utilized the author’s original codebases for these experiments, and will add these details to the appendix of the paper.
>
> **Q4) Relative Weights of Losses.**
> We did not conduct a significant hyperparameter sweep on the weighting of each of our proposed losses; rather we kept them at a ratio of 1 to 1 to 1. Due to the aforementioned constraints, we were only able to run new sweeps on toy settings and found that our approach was not sensitive to the relative weightings of the losses. It would be interesting future work to explore the precise coefficients of each individual loss, but for simplicity, we stuck with equal weightings across the losses. We will add these details to the appendix of the paper.
>
> **Q5) Empty Spectrogram Hallucinations.**
>  When no audio spectrogram is provided, even though we know that the bow is off the strings of a cello, there remains uncertainty in its precise location in the scene.Thus, GEM implicitly expresses this uncertainty by blurring over the bow’s position, and thus fails to construct the bow. In contrast, in other settings in which the audio spectrogram is non empty, there is less uncertainty over the position of the bow, as we may infer the underlying tone of the note being played, leading to a tighter distribution of possible bow positions. We will update the caption to Figure 8 with this clarification.
>
> **Q6) Nitpicks.**
> Thanks for pointing these out! We greatly appreciate your close eye for detail. We will fix all of these in the paper.

---

> > ### Comment · Reviewer_oxSK · 2021-08-24
> > **Willing to increase score**
> >
> > Thank you for the detailed response! I am currently willing to increase my score provided that all the discussed updates are incorporated in the final version of the paper (including the addition of confidence intervals for the quantitative results).

---

> > > ### Author Response · Authors · 2021-08-24
> > > **Thanks!**
> > >
> > > Yes, we will incorporate all discussed updates (including the addition of confidence intervals for the quantitative results) in the final version of the paper. Thanks for all the helpful feedback and suggestions.

---

### Official Review · Reviewer_w67Z · 2021-07-16

**Rating:** 5
**Confidence:** 4

**Summary:**

This paper proposes a modality-agnostic implicit generative model. A majority of the existing generative models involve architecture adept in a particular domain (e.g., convolutional layers for images), often requiring careful design and incurring difficulties in dealing with multi-modal data. To tackle this, the authors propose a generative manifold learning (GEM) scheme, which faithfully reflects the manifold hypothesis by utilizing hypernetwork-based implicit modeling. While implicit modeling had been proposed in previous work, unique to GEM is the use of the proposed local isometry and LLE loss functions. Such loss design removes the need of auxiliary networks (e.g., discriminator for GANs), which might require modality-dependent desgin, making the training of the model as well be completely modality-agnostic.

**Limitations And Societal Impact:**

The authors properly described the limitations and the societal impact in Sec. 6.

**Main Review:**


The paper is well organized and reads easily. The considered topic of finding a generic architecture that can be shared across different modalities is interesting and important. However, the proposed method seems to have the following weak point:

* The model seems to be good at reconstruction, but weak at generation.

Since the latent space structure/distribution is of a very flexible form, the reconstruction should be good by design. However, the cost from which is difficulty in obtaining samples from such a complicated latent distribution.

Generating samples by interpolating two points, as suggested in Sec. 4.4, could be too crude: 1) linear approximations of a manifold can have large error, especially when the number of training data is small. 2) samples obtained from such a process does not have a gurantee to converge to the true underlying density in probabilistic sense. As a result, generated samples can be inaccurate.

This might partially explain the results in Table 2 and Figure 10. The proposed model shows a FID score far worse than StyleGAN2 and the generated images are blurry. It however shows better recall, as the interpolation-based sampling scheme seems to be advantageous in covering all the data (almost perfect if reconstruction error is low).

---

All in all, it is understandable that the suggested framework and the loss functions are a solution to the modality-agnostic training; GANs need a discriminator, which usually requires modality-dependant design. However, considering the aforementioned issues, and given that the the loss functions themselves are not directly connected to the modality-independance, there seems to be an alternative method. For example, one could employ the techniques from adversarial auto-encoders (A. Makhzani et al., 2015), which can enforce the latent distribution to be Gaussian, resolving the aforementioned issues, and still be modality-agnostic.



Minor:

1) (line 278) Isn't the (non-)blurriness indeed an important property we want to measure from the generated samples?
2) The loss in Eq. (7) seems to be closely related to the Jacobian of the generative map. Is there a particular reason that Eq. (7) is used instead of Jacobian?



**Time Spent Reviewing:**

3

---

> ### Author Response · Authors · 2021-08-10
> **Reviewer w67Z Response**
>
> Thank you for your detailed comments and feedback; we appreciate the time and attention you gave to reviewing our paper. We provide high-level clarification of the underlying contributions of the paper and further provide additional comparisons and textual clarifications.  Please let us know if you have any additional questions, we are happy to clarify or provide additional experiments.
>
> ------------------------
> **Q1) Scope.**
> Please see Author Response, Section 1.1.
>
> **Q2) Generation Methodology.**
> Please see Author Response, Sections 1.2 and 2.3.
>
> **Q3) Utilizing Adversarial Loss to Enforce a Gaussian Latent Space.**
> Please see Author Response, Section 2.3.
>
> **Q4) Employing Adversarial Autoencoders.**
> We note that directly using adversarial autoencoders for multi-modal generative models is problematic, as they remain domain-dependent due to the fact that their  encoders are modality-specific (as stated in our main paper, L101-L102). For example, while a 2D convolutional network can be used to encode images, a 3D convolutional network is required to encode voxels. Furthermore, it is unclear what network architecture is needed to encode irregularly sampled data, such as that of a point cloud. In contrast, our approach enables us to encode signals independent of the data type by fitting a latent code to represent the data.
>
> **Q5) Sensitivity of FID to Blurriness.**
>  We also agree that blurriness is something we should try to prevent when generating images. However, FID is sensitive to levels of blur that are not discernible to the naked eye. For example, we find that our training image reconstructions (which obtain a MSE of approximately 0.0008 and a PSNR of 29dB, e.g.,  a recovery almost imperceptible to the human eye), obtains a FID of 23.25. The reconstructed images from VQ-VAE2 [45], which as seen in the corresponding paper are perceptually indistinguishable to ground truth images, also obtain high FID.  This is not only limited to blur, and for example, we find that simply encoding the CelebA-HQ dataset as a JPEG induces a FID of 18.19 compared to a FID of 0.0 when utilizing the original PNG encoding. These results suggest that FID may not in general be an accurate measure of visual quality, and is sensitive to blurriness or other image aberrations that are imperceptible to humans.
>
> **Q6) Jacobian of Gradient Map.**
> Yes! We could indeed replace the isometry loss with the Jacobian of the gradient map. We experimented with this idea early in the development of GEM and found that such an approach had the following weaknesses: 1) training the Jacobian of the gradient map was memory intensive, and 2) it was somewhat slow to compute the Jacobian of the gradient map itself, due to the required computation of second order gradients. In comparison, our local isometry loss is much easier to compute. We will add this remark to our Methods section, and believe this separate direction is definitely worth future exploration.

---

> ### Author Response · Authors · 2021-08-26
> **Feedback on Rebuttal**
>
> Dear Reviewer w67Z,
>
> Thank you for taking time to review and critique our paper. We were wondering if you wouldn't mind taking a look at our rebuttal response. We believe we have addressed the underlying concerns raised in the initial review. In particular, in our rebuttal response, we have clarified the main focus of our underlying paper, and have further provided both additional justifications and alternatives for sampling along the manifold, as well as comparisons towards using an adversarial latent space. We have further responded to each individual question raised in the review.
>
> Thanks,
> Paper Authors

---

### Official Review · Reviewer_C9Je · 2021-07-17

**Rating:** 6
**Confidence:** 4

**Summary:**

This paper proposes GEM model to capture the underlying structure of datasets across modalities. It can obtain perceptually consistent interpolations between samples, and can further recover points on the manifold and realize audio/image signals completion. Besides, it can also generates new samples.

**Limitations And Societal Impact:**

I don't see any potential negative societal impact of their work.


**Main Review:**

Strength:
+ The paper focuses on manifold learning and proposes an interesting idea with various interesting applications.
+ The proposed model GEM with the local isometry loss and the LLE loss outperforms the three baselines in most cases both quantitatively and qualitatively.

Weakness:
- Missing comparisons to other baselines in the literature, such as Learning Manifold Implicitly via Explicit Heat-Kernel Learning.

Question:
In the ablation study table, it shows that the local isometry loss and the LLE loss are quite effective with GEM structure. Of noting, StyleGANv2 sometimes outperforms the GEM without the local isometry loss and the LLE loss. I am wondering how these losses work with other models, say StyleGANv2?
How does the visualization in Fig.3 for GEM look like without the local isometry loss and the LLE loss?

**Time Spent Reviewing:**

2 hours

---

> ### Author Response · Authors · 2021-08-10
> **Reviewer C9Je Response**
>
> Thank you for your detailed comments and feedback; we appreciate the time and attention you gave to reviewing our paper. We address each of the raised questions on manifold comparisons, applications of manifold losses on other models, and qualitative results below, and will update the paper following each of your suggested comments. Please let us know if you have any additional questions, we are happy to clarify or provide additional experiments.
>
> ------------------------
> **Q1) Learning Manifold Implicit via Explicit Heat-Kernel Learning.**
> Please see Author Response, Section 2.1.
>
> **Q2) Manifold Losses on StyleGANv2.**
> Please see Author Response, Section 2.2.
>
> **Q3) Visualization of Figure 3 Without Losses.**
> In Figure 3 in the main paper, we find that if both LLE and Isometry losses are removed, visible artifacts can be seen in reconstructed images and other signals from the test sets. We believe this is due to the fact the implicit functions are able to overfit signals well (as seen in both NeRF (Mildenhall et al., ECCV 2020) and SIREN (Sitzmann et al., NeurIPS 2020)). We find that our proposed manifold losses regularize the autodecoder framework, ensuring that our latent space does not fall prey to overfitting.

---

> ### Author Response · Authors · 2021-08-26
> **Feedback on Rebuttal**
>
> Dear Reviewer C9Je,
>
> Thank you for all the time and effort you put into reviewing the paper. We were wondering if you could take a look at our rebuttal response. We believe that we have addressed the questions asked in the initial review. In particular, we have added a comparison to manifold learning using heat kernels, and will further add a discussion about the work in our related work. We have also added additional experimental results showing the effect of our losses of the StyleGAN2 model, finding that GEM can further improve the performance. Finally, we have answered clarification questions.
>
> Thanks,
>
> Paper Authors

---

### Author Response · Authors · 2021-08-10
**Author Response (1/2)**

We thank reviewers for their detailed comments and feedback. We’re glad that reviewers unanimously agree that Generative Manifold Learning (GEM), is an interesting and different research direction (“an interesting idea with various interesting applications” (C9Je), “interesting and important” (w67Z), “novel approach to manifold learning” (oxSk)). Reviewers thought the paper was well-written (“well organized and reads easily” (w67Z) and “organized and written clearly” (oxSk)), and believed that the claims of the paper were also well-validated (“evaluation protocol is extensive and is both qualitative and quantitative” (oxSk)).

However, reviewers were concerned about our comparisons with related work. In particular, reviewers desired comparisons against: 1) the framework proposed in  “Learning Manifold Implicitly via Explicit Heat-Kernel Learning” (C9Je), and 2) a model which utilizes  a Gaussian latent space via “adversarial autoencoders” (w67Z). Further, reviewers had questions about the specific implementation details of - and design choices underlying - the work, such as how the hyperparameters for each of the individual baselines were chosen (oxSk). Reviewer (w67Z) additionally voiced concerns about the generative capability of GEM. Finally, we address remaining concerns posed by each reviewer in the individual author responses below.

We believe that we have been able to address all concerns of the reviewers with our additional experiments (detailed below), coupled with our per-reviewer replies.

--------------------------

## 1. Clarifications

### 1.1. Scope.
Reviewer w67Z points out that the generative performance of GEM is not on par with state-of-the-art generative adversarial approaches. We would like to clarify that the primary focus of our paper is **not** generative modeling, but rather, manifold learning, i.e., the discovery of the “structure of datasets across modalities”. While we agree that the generative performance of GANs is impressive, they fall short in the other aspects which we  discuss, most importantly, coverage of the **full** manifold of data. As the discriminator is modality-dependent, there is further no straightforward way to apply GANs to datasets other than images. With GEM, we propose an approach to recovering the low-dimensional subspace in which high-dimensional perceptual signals **of any modality** lie. Progress in this direction would enable new applications, some of which we prototype in the paper, such as cross-modal completion, joint learning of structured embedding spaces across modalities, etc. We further extensively validate the ability of GEM to capture the underlying structure of multi-modal data in section 5.2 via interpolations, inpainting, multimodal completion, t-SNE embeddings, and nearest neighbor embeddings. Across these applications, we show that the latent space of GEM not only covers the full data manifold and discovers the structure of the underlying data, but also demonstrate that GEM outperforms generative adversarial models. Finally, we offer generative modeling only as an additional experiment, but note that this is **not** the core contribution. We believe that this may nevertheless inspire follow-up work on this new approach to modality-agnostic generative modeling. We will clarify this in the paper, highlighting that we do **not** claim to outperform GANs at generation, that improving the generative ability of GEMs is an exciting avenue for future research, and that our core objective is manifold learning.

### 1.2. Generation Methodology.
While the generative performance of GEM is not on par with state-of-the-art generative adversarial approaches on images, we believe that our results validate that GEM is an exciting first step towards a manifold-learning approach to generative modeling, distinct from both existing GANs and VAEs, and, therefore, opens up interesting avenues for future work. Furthermore, we note that on multimodal domains, our approach does obtain the best performance compared to all other approaches.

Reviewer w672 is concerned that our proposed sampling methodology is unlikely to sample from the true underlying manifold. Non-parameteric manifold learning has had a long history in machine learning [2, 3, 7-10], with GEM being a particular instance of this framework. The typical approach to sampling from such manifolds is traversal of nearest neighbors [2], since a manifold by definition is assumed to be locally euclidean around each training point. Our approach, through the use of noisy nearest neighbor interpolation, can be seen as a variant of this classical scheme. To further ensure that such a sampling scheme works well, during training, we leverage the LLE loss to explicitly enforce that our manifold locally resembles convex linear subregions R_i (main paper L149-L151). While a formal proof is outstanding, we argue that if our method succeeds in discovering the true manifold leveraging the LLE loss, sampling by interpolation consequently also samples from the true manifold.

Nevertheless, we note that our current approach may face particular challenges when asked to interpolate points that are largely disconnected from other points on our manifold (though other models such as GANs will instead choose to ignore such points). A potential remedy for this issue is to learn a generative model directly over the auto-decoded latents of training data points which define our manifold. Furthermore, assuming an infinite number of training data points, and thus latents, we have the guarantee that a maximum likelihood model would probabilistically sample from the underlying data distribution. We report results using this in Section 2.4.

Next we provide a substantial number of new experiments addressing the concerns raised by reviewers regarding additional model comparisons, choice of hyperparameters, and confidence intervals of results.

--------------------------

## 2. Additional experiments - !! please see per-reviewer responses for more reviewer-specific experiments !!

### 2.1 Comparison with Learning Manifold Implicitly via Explicit Heat-Kernel Learning.
We compare GEM against the method described in “Learning Manifold Implicit via Explicit Heat-Kernel Learning” on our task of test set reconstruction. We utilize the underlying learned heat kernel to reconstruct test images in CelebA. To reconstruct a test image, we compute the kernel weights of all training images with respect to said test image, and reconstruct the test image using the corresponding weighted sum of training images. We find that the resulting reconstruction yields a MSE of 0.0365 and a PSNR of 14.99, which is similar in performance to a VAE, but worse than GEM. We will add these results to Figure 3. Additionally, we will include a discussion of the heat kernel implicit manifold learning technique to our related work on, where we will compare and contrast the method with that of GEM.

### 2.2 Utilizing Manifold Losses on Other Models.
Reviewer C9Je further asked if our proposed manifold losses may also be utilized on other models. We find that incorporating our manifold losses into other models also boosts their performance. Specifically, we test our losses on the StyleGAN2 model. On our test reconstruction task (over CelebA images), we find that using only the StyleGANv2 objective gives a MSE of 0.0044 and a PSNR  of 24.03. The addition of auto-decoding with the StyleGANv2 objective elicits a MSE of 0.0041 and a PSNR of 24.29. The further incorporation of our LLE loss advances the reconstruction performance to a MSE of 0.0038 and a PSNR of 24.61. We note that we do not find any additional boosts to the MSE reconstruction by including the isometric loss, perhaps due to the fact that the underlying latent space is Gaussian. We will add these results to the main paper.

---

> ### Author Response · Authors · 2021-08-10
> **Author Response (2/2)**
>
> ### 2.3 Utilizing Adversarial Loss to Enforce a  Gaussian Latent Space.
> Reviewer w67Z asked if an alternative method for signal-agnostic generative modeling is to enforce that the underlying latent space follows a Gaussian distribution by leveraging an “adversarial autoencoder”.  While we agree that this is an alternative method for generative modeling, early in our experimental design, we explored this idea -- wherein we enforced that the latent space had a low KL with respect to a normal distribution. Unfortunately, we found that despite this objective, at test time when we generated new samples by sampling from a normal distribution, we generated significant noise in our samples. Concurrent to our work, [29] also found a similar effect. In the rebuttal period, we further experimented with deploying an adversarial loss to ensure that our latent space was Gaussian. However, we uncovered that this again resulted in significantly noisier samples, with samples obtaining an FID of 114.55. We discuss this in the method section of the paper.
>
> ### 2.4 Sampling on the Manifold Utilizing a Generative Model.
> Reviewer w67Z further asked if there was an alternative method to sampling from our manifold: one which may not have difficulty in areas of low data density. To address this, we explore alternatively sampling from the manifold discovered by GEM by utilizing a generative model, in particular a GAN. We directly train a GAN, using 1) auto-decoded latents of each training data point (similar to [1]) as positive samples to a discriminator, and 2) random samples from a normal distribution as negative samples to a discriminator. We find that such a GAN model on the CelebA dataset obtains a FID of 38.96, comparable to the results we obtained through interpolation. This indicates that samples generated through interpolation lie on the underlying manifold to a similar extent to those discovered by the GAN. We will add these additional experiments to the main body of the paper.
>
>
>
> ### 2.5 Variations Across Seeds.
> Reviewer oxSk asked about the confidence intervals underlying our reported evaluation metrics. To explore the variability of our results, we reran the test time reconstruction experiments (whose results we had we showed Figure 3) over three separate seeds. We report results on test image reconstruction of CelebA in the table below. We find that our approach outperforms all alternative models considered -- even with confidence error bounds, we do not find overlapping intervals. We will update the quantitative results in paper with error bounds in the camera ready version.
>
> | Method      | MSE  | PSNR |
> | ----------- | ----------- | ----------- |
> | VAE    | 0.0327 +/- 0.0035     | 15.16 +/- 0.06      |
> | FDN   |   0.0062 +/- 0.0003       | 22.57 +/- 0.02      |
> | StyleGANv2   |   0.0044 +/- 0.0001     | 24.03 +/- 0.01       |
> | GEM  |   0.0025 +/- 0.0001       | 26.53 +/- 0.01       |
>
>
>
>
> [1] Achlioptas, Panos, et al. "Learning representations and generative models for 3d point clouds." International conference on machine learning. PMLR, 2018.

---

### Decision · Program_Chairs · 2021-09-27

**Decision:**

Accept (Poster)

**Comment:**

The paper proposes GEM, an approach for multi-modal manifold learning, which is novel and with interesting results. I found the rebuttal very well done and I think it clarified many open issues with the approach, although not to the degree to have a total agreement of the reviewers. Nevertheless I find the clarifications sufficient and I am inclined to accept the work.